# Great Lakes Waves Forecast System on High-Resolution Unstructured Meshes

Ali Abdolali[1,2], Saeideh Banihashemi[3,4], Jose Henrique Alves[5], Aron Roland[6], Tyler J. Hesser[1], Mary Anderson Bryant[1], and Jane McKee Smith[1]

[1]US Army Engineer Research and Development Center, Coastal and Hydraulics Laboratory, Vicksburg, MS, USA
[2]Earth System Science Interdisciplinary Center (ESSIC), College Park, MD, USA
[3]NWS/NCEP/Environmental Modeling Center, National Oceanic and Atmospheric Administration (NOAA), College Park, MD, USA
[4]Lynker, Leesburg, VA, USA
[5]Earth Prediction Innovation Center, NOAA/Oceanic and Atmospheric Research/Weather Program Office, Silver Spring, MD, USA
[6]BGS IT & E, Darmstadt, Germany

**Correspondence:** Ali Abdolali (ali.abdolali@usace.army.mil)

**Abstract.** Wind-wave forecasts play a crucial role in the North American Great Lakes region towards ensuring the safety of communities, enhancement of the economy and protection of property. Modeling wind waves in closed and relatively shallow basins with complex bathymetry like the Great Lakes is a challenge that is successfully tackled in part by using variable-resolution triangular unstructured meshes with no limits in terms of computational scalability and maximum resolution in the coastal areas. In this paper, we discuss recent advances in developing unstructured-mesh capabilities as part of the spectral wave model WAVEWATCH III, in the context of National Oceanic and Atmospheric Administration (NOAA) operational requirements such as model robustness, efficiency, and accuracy. We revisit the history of developments leading to the transition from rectilinear, to curvilinear grids, and finally to an unstructured mesh version of NOAA's operational Great Lakes wave-modeling system (GLWUv2.0). The article describes the development of the operational GLWUv2.0, from mesh design and scalability analysis to validation and verification for hindcast of storm cases and reforecast using 4 months of retrospective simulations. In closed Great Lakes basins untouched by swell from distant sources, the atmospheric model's direct impact on wave behavior stands apart, showing reduced forecast accuracy over time, while maintaining consistent precision in accurately wind-hindcasted stormy conditions.

The North American Great Lakes play a crucial role in the social, economic, and environmental fabric of the United States and Canada, supporting the livelihoods and well-being of millions of people. The region is home to approximately 8% of the U.S. population (eight U.S. states of Illinois, Indiana, Michigan, Minnesota, New York, Ohio, Pennsylvania, and Wisconsin) and 32% of Canada population (two Canadian provinces of Ontario and Quebec). Local communities rely on the Great Lakes for various purposes, including drinking water, industrial and agricultural activities, transportation, and recreation. The region further supports a range of economic sectors, including manufacturing, tourism, fishing, and shipping. Great Lakes waterways enable the transportation of goods and raw materials, supporting trade and commerce within the region and beyond.

Accurate wind wave forecasts play a crucial role in ensuring the safety of Great Lakes communities, protecting coastal properties, and facilitating smooth maritime operations. As described in Alves et al. (2014) and Alves et al. (2023), the history of forecasting in the Great Lakes region dates back to the establishment of NOAA's Great Lakes Environmental Research Laboratory (GLERL) in 1974, which marked the beginning of systematic marine forecasting in this area. In collaboration with researchers from the Ohio State University, GLERL pioneered the development of the first wave forecasting system, based on a parametric, first-generation wave model, specifically tailored for the Great Lakes during the early 1980s (Schwab et al., 1984). As advancements in wind-wave modeling progressed and third-generation models emerged in the late 1980s, GLERL, in partnership with NOAA's Environmental Modeling Center (EMC), successfully co-developed a next-generation forecast system. This system was integrated into NOAA's operational wave model framework, becoming an integral part of the suite of operational environmental forecast models at the National Centers for Environmental Prediction (NCEP).

Upon the release of the third generation wave model WAVEWATCH III (Tolman, 1999) at EMC, GLERL and NCEP initiated the development of the next generation of wave forecast system for the Great Lakes Waves (GLW) in 2004, incorporating key requirements from forecasters and science operations officers (SOOs) from regional Weather Forecast Offices (WFOs). Two years later in 2006, the first WW3-based GLW system was implemented operationally with three-hourly intervals. The model grid was a single rectilinear grid with a resolution of $\sim 4$ km, with coverage of all five major Great Lakes basins (Erie–Saint Clair, Ontario, Huron, Michigan, and Superior). Details are provided in Alves et al. (2014).

The next upgrade featuring a 2.5-km Lambert-conformal spatial grid took place in 2015, enhancing the forecast due to a better representation of the basin-wide geographical features, particularly during rapidly changing conditions (Alves and Chawla, 2015). This upgrade shed light on the importance of grid resolution, and the need for the implementation of more accurate physics, especially in coastal environments (Alves et al., 2023). However, the existing system was not sufficiently computationally efficient to resolve coastal areas with a uniform resolution grid. Meantime, the core WW3 model was equipped with unstructured mesh capabilities and more advanced nearshore physical parameterizations including, but not limited to, depth breaking, triad interaction, and reflection (Abdolali et al., 2020). The use of a single triangular mesh from coarse offshore to fine nearshore resolution instead of multiple inter-nested grids with different resolutions showed a substantial improvement in the workflow design, maintenance, computational efficiency, and accuracy.

In 2017, the GLW system was upgraded to an unstructured mesh (GLWUv1.0), with a 2.5 km resolution in deep water down to 250 m in the nearshore. The added benefit as a result of better representation of wave transformation in the coastal areas was recognized by the forecasters in the regions within this implementation (Alves et al., 2023). GLWUv1.0 used the explicit numerical solver and the same parallelization algorithm as the structured and curvilinear grids, mainly due to operational limitations. The approach had several limitations including restrictions on time stepping due to resolution and grid size, and a maximum number of CPU threads dependent on the number of spectral components. With the advent of a new parallelization algorithm and an implicit numerical solver, the model was made more efficient and accurate on very large and fine resolution meshes. The contrast lies in how Domain Decomposition ($DD$) surpasses Card-Deck ($CD$) in scalability with a large number of CPUs. $CD$ has a restricted maximum count of CPUs compared to the unlimited count in $DD$. Moreover, when dealing with

finer resolution meshes, the implicit solver in $DD$ enables operation with larger CFL numbers, whereas the explicit solver in $CD$ is limited by $CFL < 1$, resulting in slower model performance. (Abdolali et al., 2020).

With the availability of appropriate resources, a recent GLWU implementation enabled taking advantage of such features, with enhanced benefits that will be discussed below, providing a unique opportunity to improve the representation of nearshore wave transformation, incorporating water level and current effects and resolving complicated geometries in shallow water regions (Moghimi et al., 2020). In the past few decades, the noticeable increase in frequency and destructiveness of coastal storms has led to a growing recognition of the need to couple wave models with other earth system models. This necessity extends beyond just public awareness and warning systems, and its benefits encompass applications such as hindcasts for climatologies, risk analyses for the insurance industry, and relevant data for tourism, investment sectors, and numerous other stakeholders. The coupling of atmospheric, ocean wave, surge, and hydrological models on high-resolution numerical grids has improved model accuracy by better representing nearshore/inland geometries and physics (Moghimi et al., 2020).This breakthrough has expanded the limits of WW3 to be dynamically coupled with storm surge, hydrological, ice, and atmospheric models and provides the opportunity to investigate nearshore wave climate (Bakhtyar et al., 2020). Moreover, the highly efficient WW3 model allowed researchers to evaluate uncertainty via ensemble modeling (Abdolali et al., 2021).

This paper focuses on the latest operational implementation of the Great Lakes wave model GLWUv2.0, but also discusses new features added to the wave model that benefit earth systems coupling and ensemble forecasting. The sections are arranged as follows. Section 1 provides an overview of the GLWUv2.0 implementation in operation in May 2023, including mesh design, forcing characteristics, the end-to-end workflow design based on the explicit solver in WW3, retrospective simulations for two summer and two winter months, and validation analysis. In Section 2, the performance of the unstructured WW3 was analyzed for ten hindcasted windy conditions over the Great Lakes, highlighting the scalability of the domain decomposition algorithm and the effectiveness of the implicit solver on high-resolution meshes. The contrasts, shown in Sections 1 and 2, manifest the lag between operation and core model development front and define the path forward in the future of operational forecast. Concluding remarks and requirements for future validation studies and support of better wave forecasts, in particular in the winter season and coastal areas, are provided in Section 3.

## 1    Operational Implementation

The Great Lakes Wave Model (GLWUv2.0) currently provides guidance to twelve Weather Forecast Offices (WFOs) on six domains, including the five Great Lakes region plus Lake Champlain. The system is forced by NOAA operational wind sources, as described in Section 1.2. The Great Lakes wave forecast operates on an hourly basis, incorporating both short and long cycles, as depicted in Figure 1. There are twenty short cycles, each running for 48 forecast hours. Additionally, there are four long cycles, which extend for 150 forecast hours and are scheduled at $01z$, $07z$, $13z$, and $19z$. The core model, WAVEWATCH III, takes advantage of key features from the most recent advancements in the WW3v7 model package outlined above, with some restrictions due to NOAA operational requirements (NCO, 2022), such as end-to-end run-time, output formats, and

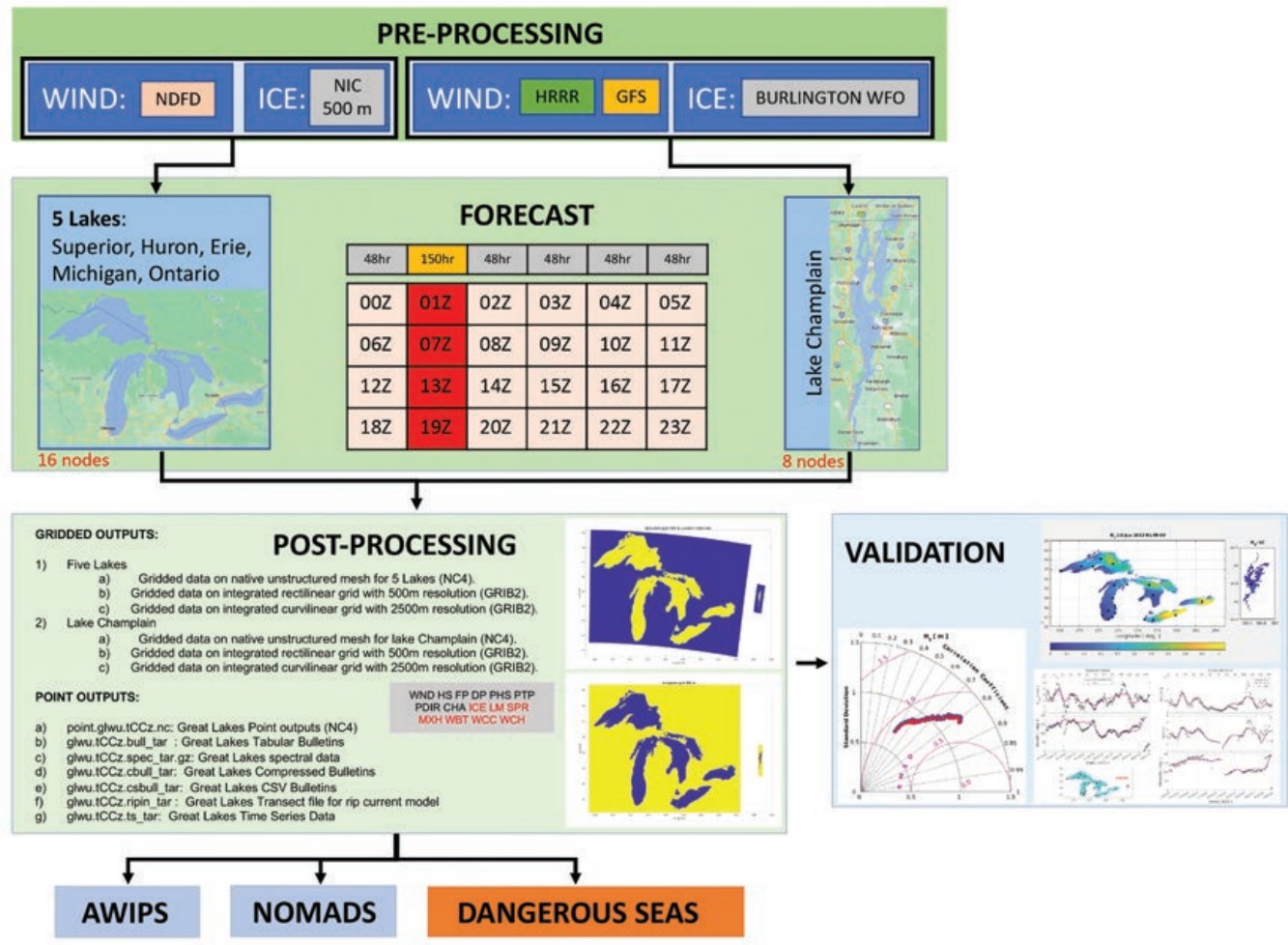

**Figure 1.** Great Lake Wave Unstructured v2.0 workflow at NOAA NCEP Central Operation (NCO).

Operational Readiness Test (ORT). These restrictions mandate the use of the explicit solver for the GLWUv2.0 implementation without incorporating water level and current velocity.

The GLWUv2.0 model resolves the wave energy density spectrum with frequencies between 0.05 and 1.055 Hz, divided
90 into 32 frequency bands with a geometric increment factor of 1.1, and 36 directions with $10°$ increments. In addition, wave physics parameterizations include the Ardhuin et al. (2010) wind input and wave dissipation source-term (ST4), the Generalized Multiple Discrete Interactions Approximation (GMD) of Tolman (2014) for nonlinear wave-wave interaction (NL3, replacing the Discrete Interaction Approximation (DIA) of Hasselmann et al. (1985)), JONSWAP bottom friction (BT1; Hasselmann et al., 1973), and depth-limited breaking based on the Battjes-Janssen formulation (DB1) (Battjes and Janssen, 1978). In
95 order to take into account ice, a simple ice blocking parameterization (IC0) with the discontinuous method is used, where a critical ice concentration in which the scheme switches between free propagation and blocking is $\epsilon_{c,0} = \epsilon_{c,n} = 0.7$. Due

to NOAA operational requirements, a parallelization scheme known as "Card Deck" and the explicit numerical scheme are utilized, which limits performance due to restrictions on the number of allowed parallel threads, dependent on the discrete wave spectrum resolution, and on time stepping. In view of the latter, the global, spatial, spectral, and minimum source term WW3 time steps are set to 180, 60, 90 and 10 s, respectively.

## 1.1 Unstructured Mesh

In the GLWUv2.0, the $G0$ unstructured mesh is utilized for the five lakes with $\sim 253k$ nodes and $\sim 418k$ elements. A new feature in the latest implementation is the addition of a Lake Champlain mesh containing $\sim 30k$ nodes and $\sim 62k$ triangular elements, ranging in size from approximately $60\ m$ in dynamic coastal regions to approximately $400\ m$ in less dynamic offshore areas. The Great Lakes Bathymetry collection compiles geological and geophysical data of five lake floors, including bathymetry and detailed maps sourced from over a century of soundings by various organizations like the U.S. Army Corps of Engineers, NOAA, and the Canadian Hydrographic Service. NCEI/NOAA compiled unified topobathy data for Lake Erie and Saint Claire (NGDC, 1999a), Lake Huron (NGDC, 1999b), Lake Michigan (NGDC, 1996), Lake Ontario (NGDC, 1999c), and Lake Superior and provided public access to this data. The topobathymetric grid for the generation of the Lake Champlain mesh was obtained by refining the Environment and Climate Change Canada (ECCC)-developed mesh, integrating data from 15 sources for a detailed two-dimensional hydrodynamic model. Covering from South Bay to the Poultney River in Whitehall, NY, and extending northward to Fryers Rapids near St Jean, QC, it intricately maps 14 significant river inputs to the lake. This grid encompasses surrounding floodplains to simulate various inundation scenarios across the spectrum of water level fluctuations experienced within the region (Titze et al., 2023). Mesh resolution and corresponding histograms, highlighting the distribution of element size and significance of coastal elements in comparison to deep-water elements, are shown in Figure 2, and the summary is provided in table 1. Mesh systems were tested to ensure they resolved key morphological features in all domains while remaining computationally feasible in a real-time operational environment.

| Lake | # Node | # Element | $\Delta x_{min}$ (m) | $\Delta x_{max}$ (m) |
|---|---|---|---|---|
| Superior | 51k | 81k | 246 | 3300 |
| Michigan | 58k | 103k | 250 | 2470 |
| Huron | 64k | 101k | 203 | 2840 |
| Erie | 45k | 78k | 203 | 1603 |
| Ontario | 35k | 57k | 224 | 2150 |
| Champlain | 30k | 60k | 60 | 400 |

**Table 1.** Mesh characteristics for Lakes Superior, Michigan, Huron, Erie, Ontario and Champlain in terms of number of nodes and elements, minimum and maximum resolutions.

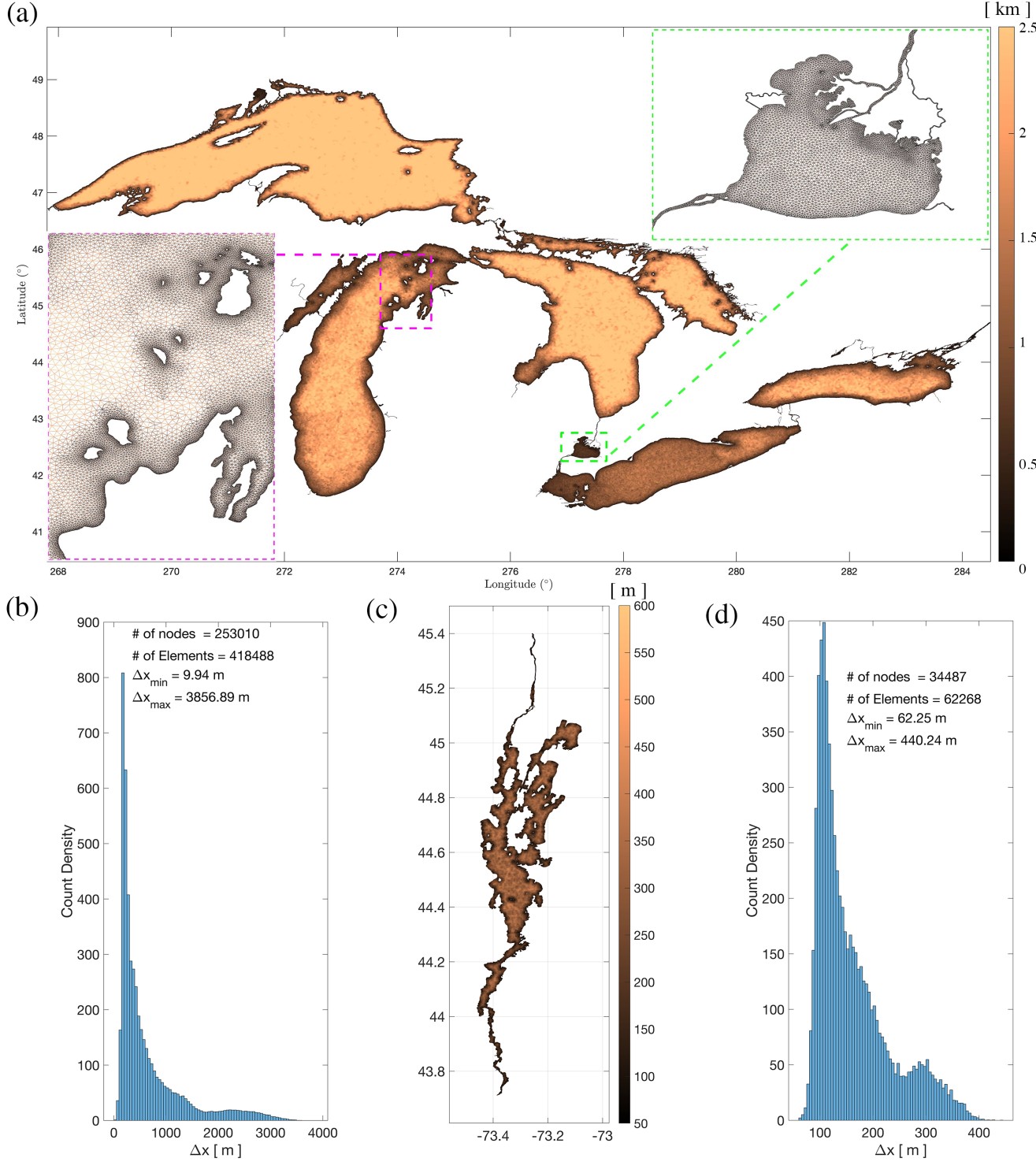

**Figure 2.** Great Lake Wave Unstructured v2.0 mesh resolutions: (a,b) five Great Lakes and (c,d) Lake Champlain.

## 1.2 Forcing Fields

The operational GLWUv2.0 forcing includes temporally variable wind speed and a stationary ice concentration, defined at the initialization time step and kept constant for the entire cycle. A combination of various sources is used for wind due to operational limitations such as availability and spatial and temporal coverage. A flowchart of the forcing fields is shown in Figure 3.

The wind for Lakes Superior, Michigan, Huron, Erie, and Ontario are extracted from the National Digital Forecast Database (NDFD) with a spatial resolution of $2.5 \ km$ and variable temporal resolution up to 150-hour forecast (1-hourly out to 24 hours, 3-hourly for days 2-3, 6-hourly for days 4-6). The NDFD is a combination of data from regional NWS Weather Forecast Offices (WFOs) and the National Centers for Environmental Prediction (NCEP) models (Glahn and Ruth, 2003). The corresponding resolution of the sea ice concentration analysis for these five domains is $500 \ m$ (as opposed to $5 \ km$ in v1.0), obtained from the National Ice Center (NIC). The NIC Data for the Great Lakes are created from daily ice analysis. The files contain information on ice conditions that are separated into total ice concentration, ice types with their respective concentrations, and ice floe size.

The Lake Champlain portion of the system is forced with atmospheric wind data from NCEP's High-Resolution Rapid Refresh (HRRR) with 3-$km$ resolution for the first two days and switched to Global Forecast System (GFS) with 0.25 degrees ($\sim$ 27-$km$) resolution up to 150 hrs. The HRRR is a NOAA real-time 3-km resolution, hourly updated, cloud-resolving, convection-allowing atmospheric model, initialized by $3 \ km$ grids with $3 \ km$ radar assimilation. Radar data is assimilated in the HRRR every 15 min over a 1-hr period adding further detail to that provided by the hourly data assimilation from the 13 $km$ radar-enhanced Rapid Refresh (Dowell et al., 2022). The Global Forecast System (GFSv16) from the National Centers for Environmental Prediction (NCEP) serves as a fundamental component in NCEP's operational numerical guidance suite for global climate modeling (NOAA, 2021). It offers both deterministic and probabilistic global forecasts, extending up to 16 days, and plays a key role by supplying initial and boundary conditions for NCEP's regional, ocean, and wave prediction models. The Lake Champlain ice coverage data is taken from the National Weather Service Weather Forecast Office in Burlington, VT.

## 1.3 Workflow

The main characteristics of an end-to-end workflow in an operational environment are its robustness, predictability of potential unavailability of inputs with replaceable alternative options, and full automation, with no human intervention and following a strict run-time scheduling to ensure on-time delivery of guidance products. The three main steps of the GLWUv2.0 workflow, as shown in Fig. 1, are pre-processing, forecast, and post-processing jobs for two parallel jobs, one for the five Great Lakes and the other for Lake Champlain. The workflow runs every hour for 48 hrs in the short cycles and 150 hrs in long cycles, four times a day (01z, 07z, 13z, and 19z). Within the Great Lakes domain, the run-time is 24 minutes for the short cycle and 50 minutes for the long cycle. For Lake Champlain, the process completes in 17 minutes in the short cycle and 27 minutes in the long cycle.

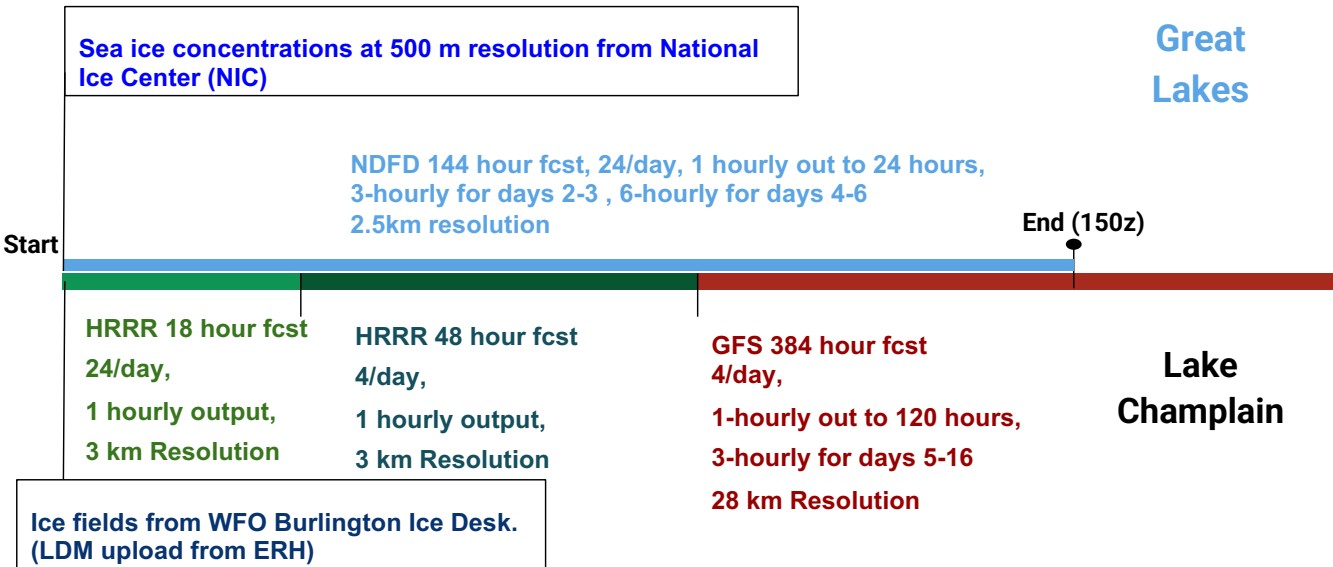

**Figure 3.** Great Lake Wave Unstructured v2.0 atmospheric and ice forcing hierarchy. For wind, the National Digital Forecast Database (NDFD) and a combination of High-Resolution Rapid Refresh (HRRR) and Global Forecast System (GFSv16) are used for the five lakes and Lake Champlain, respectively. The ice is taken from the National Ice Center (NIC) and WFO Burlington help-desk for the five lakes and Lake Champlain, respectively.

### 1.3.1 Pre-processing job

This serial job retrieves wind and ice inputs from other operational models like HRRR, GFS or analysis from NDFD, NIC, and the Burlington Weather Forecast Office. In case the forcing for the current cycle is not available, a look back option fills the forcing from the previous forecast cycles. If the ice field is not provided, the previous forecast cycle ice field is used. It is worth noting that the ice field is constant during the entire forecast job and initiated at the beginning of simulation. Within this step, the grib2 files are cropped for GLWU coverage, interpolated on the computational grid, and saved in NetCDF format. The

WW3 pre-processing executable ($ww3\_prnc$) converts the inputs into binary format. The run-times for pre-processing step are 3 and 4 minutes for short and long cycles for the five lakes and 8 and 13 minutes for short and long cycles for Lake Champlain, respectively.

### 1.3.2 Forecast job

This parallel job performs two simulations for the five lakes (on 16 nodes, each node with 64 cores) and Lake Champlain (on

8 nodes) simultaneously. Note that the reason for the separation into two domains is to load balance between the two domains in which Lake Champlain requires more iterations (due to CFL criteria). The run-time for the five lakes is 9 and 24 minutes for

short and long cycles, while Lake Champlain takes 5 and 10 minutes, respectively. Binary point and gridded outputs are stored during the simulation.

### 1.3.3   Post-processing job

This serial job generates NetCDF and grib2 outputs on 500 $m$ and 2.5 $km$ resolutions while point outputs are generated in NetCDF format (spectral outputs and time series of wave statistics). These data are transferred to the Advanced Weather Interactive Processing System (AWIPS) and NOAA Operational Model Archive and Distribution System (NOMAD) for use by WFOs and the public. The runtime of this step is 12 and 22 minutes for short and long cycles for the five lakes and 3 minutes for both short and long cycles for the Lake Champlain.

### 1.3.4   Validation and verification job

Once the field and point outputs are prepared during the post-processing phase, and real-time observations from buoys are gathered and subjected to quality checks, a comparative analysis of the results is conducted. This analysis involves generating statistical summaries, including time series plots for parameters such as wind speed, wind direction, significant wave height, peak period, and mean wave direction. Additionally, graphical representations such as linear regression plots and Taylor dia-
grams are generated, along with tables containing relevant statistics.

### 1.4   Validation and Verification

The long cycle runs with 6-day re-forecasts simulations (four times/day) for the GLWUv2.0 system are validated against in-situ measurements at 25 locations, shown in Figure 4. The model performance has been evaluated during two summer months (June and July 2022) where observations were available. Note that the temporarily deployed buoy in Lake Champlain was available
during this period, which was the criteria for the selection of the retrospective summertime run period.

Two months of simulations (January and February 2022) were conducted to evaluate the model results that included the higher resolution ice and assess model performance with previously reported observed wave artifacts at the ice edge by WFOs (Figure 5). The buoys are removed during wintertime to avoid damage to the gauges, therefore, the forecasters qualitatively verified agreement of the model with their oberservations.

The physical parameters in the wave model were tuned to minimize the statistical metrics of significant wave height, keeping the first forecast day close to the observations. As shown in Fig. 6, the model score in terms of bias, absolute bias, standard deviation, and root mean square error deteriorate with longer forecast lead time, which introduced less accuracy and more uncertainty in the forcing field.

Looking closely at trend between forcing wind and downstream wave model output in Fig. 7, the linear regression plot is
shown for results separated for percentile ranges of <50, 75, 90, 95 and > 99 for up to six daily forecast. It is clearly shown the wind ($U_{10}$) was overestimated in NDFD for small values (< 50 percentile) and then underestimated each day for larger values since the forecast was initiated. On the other hand, the wave model outputs were close to 1:1 for the first day with slight

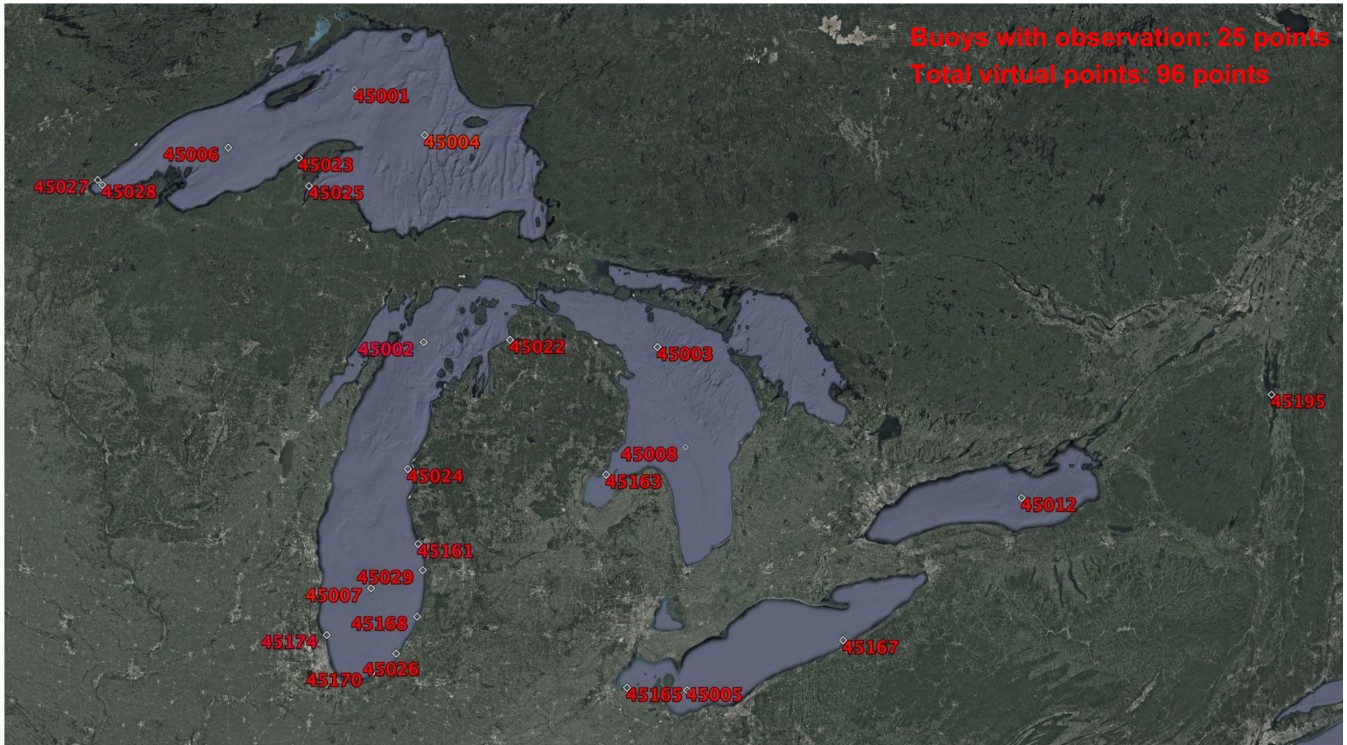

**Figure 4.** The location of meteorological and wave observations, obtained from the National Data Buoy Center (NDBC) locations for model validation (Google Earth, earth.google.com/web/)

underestimation for the larger waves (panel $b$). Note that unless a dynamic forecast lead time dependent correction is applied on the wind field, the wave models cannot be tuned for the whole duration of the forecast, and the results would deviate from
observations as forecast lead time increases. The pattern propagation (under/overestimation) from wind to wave model is due to the nature of wind-wave generation and the direct impact of wind on waves.

Last, the performance of the NDFD winds and WW3 waves at 25 in-situ observations were assessed as a function of forecast lead time and summarized in the Taylor diagrams shown in Fig. 8, in terms of normalized standard deviation ($\sigma$), root mean square deviation ($RMSD$), and correlation coefficient ($CC$). For $U_{10}$ (top panel), as forecast lead time passes, the $RMSD$
increases from 0.7 to 0.9 while $\sigma$ and $CC$ drop from 0.98 and 0.74 to 0.5 and 0.32, respectively. On the other hand, a similar pattern is observed for $H_s$ (bottom panel), with a slight change in $RMSD$ (0.7-0.8) whereas $\sigma$ and $CC$ drop from 1.23 and 0.8 to 0.7 and 0.5, respectively.

## 2  Future Implementations

In this section, key features of the core WW3 model left out from GLWUv2.0 due to operational restrictions are tested in
support of future upgrades. The aim is to highlight the performance and efficiency benefits of new model features in meshes

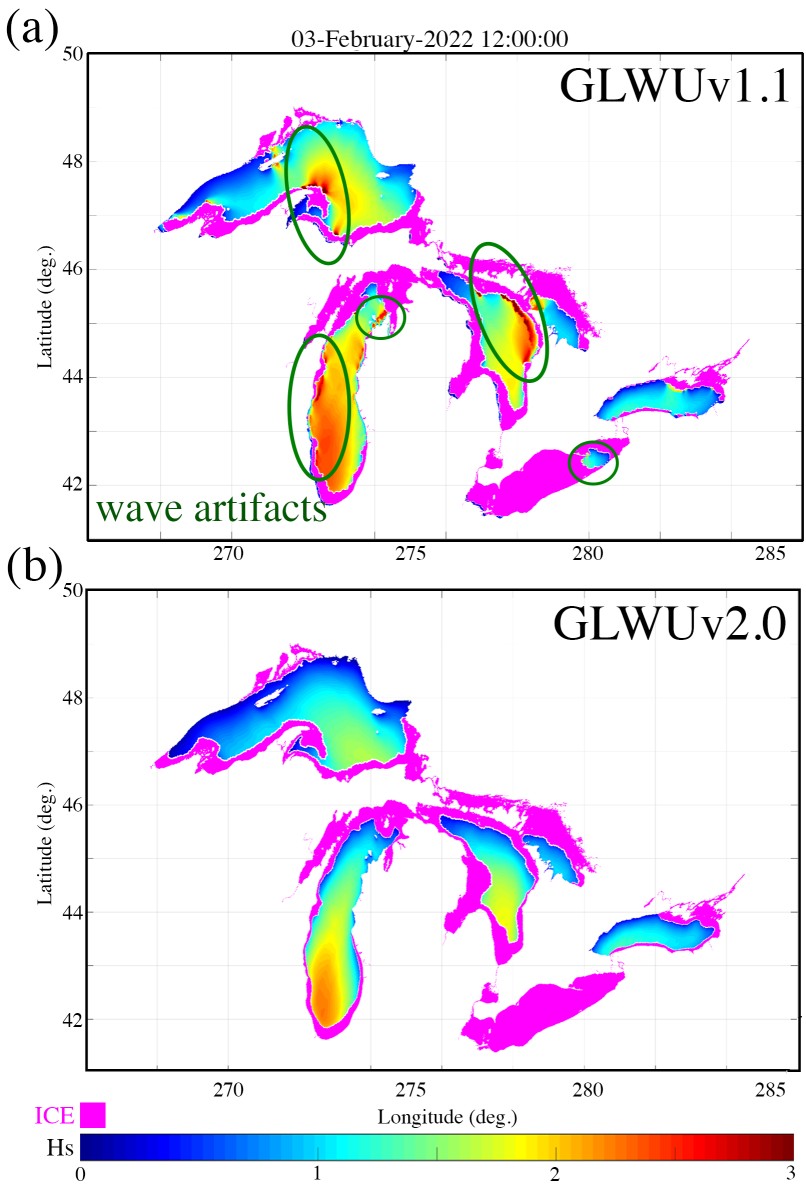

**Figure 5.** Comparison of the GLWU domain significant wave height between the GLWUv1.1 (a) and the development GLWUv2.0 (b) during ice season (a snapshot on February 3rd, 2022 12:00:00 EST). The green circles in panel (a) show the wave artifacts along ice edges.

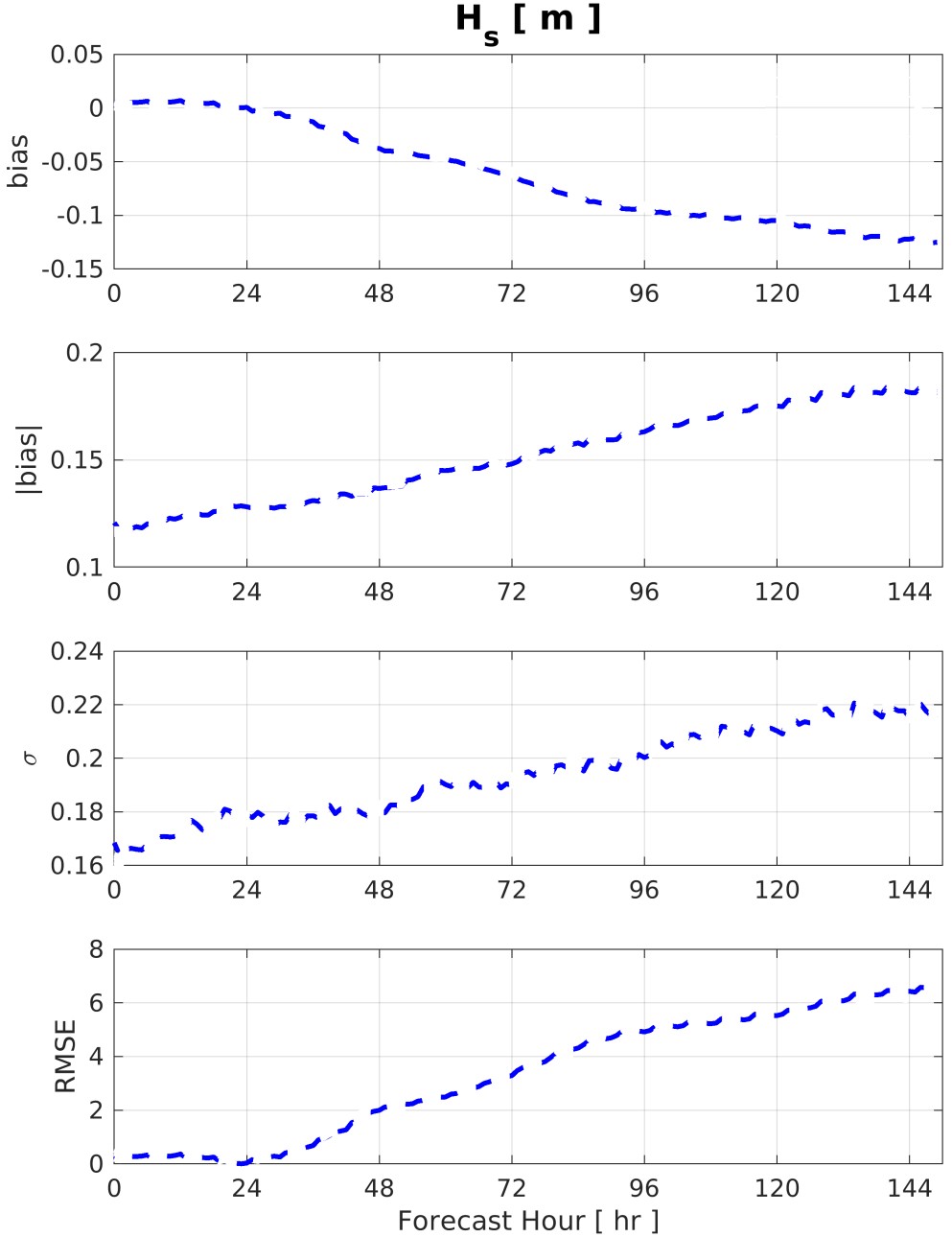

**Figure 6.** Wave model performance for hourly $H_s$ forecast versus lead time in terms of bias, absolute bias, standard deviation ($\sigma$), and root mean square error ($RMSE$).

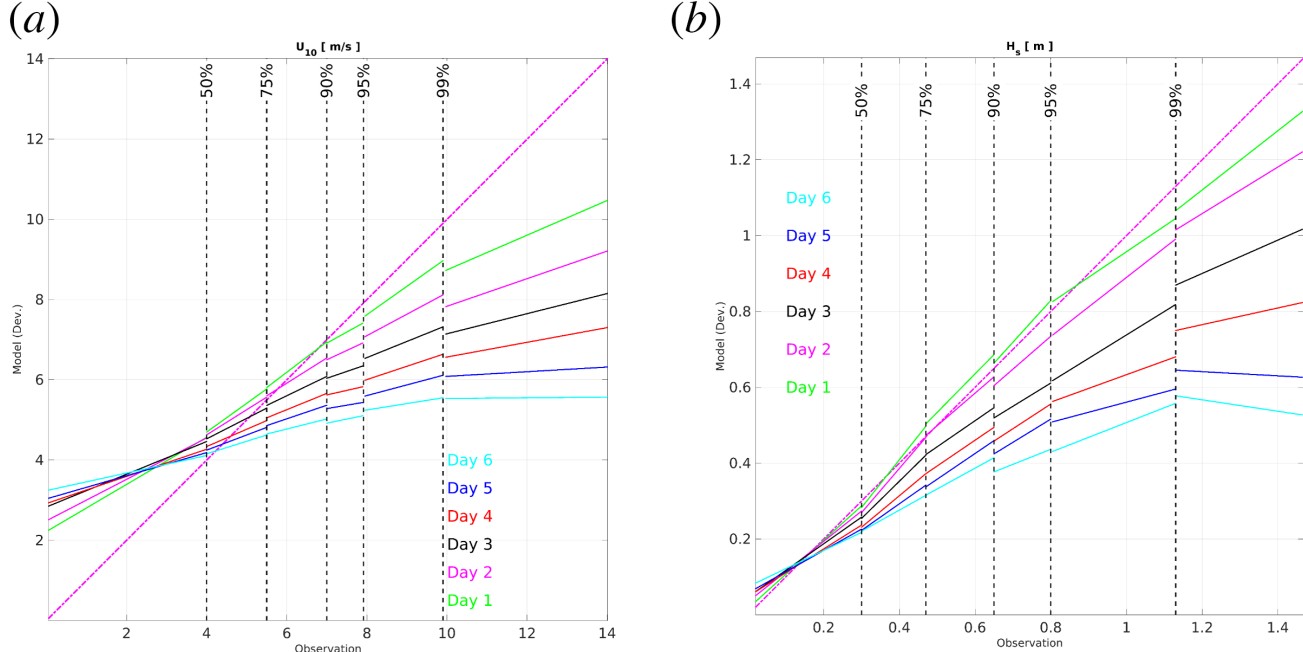

**Figure 7.** Linear regression plot for wind speed (a) and significant wave height (b) for the daily forecast lead times, separated for <50, 75, 90, 95 and > 99 percentiles.

of one order of magnitude larger than the one used in operation ($G0$) with high resolution ($\sim$ 5-10 m) in coastal regions. The model configuration for simulations closely resembles those outlined in Section 1, differing in the following aspects: the parallelization algorithm utilized here is Domain Decomposition instead of Card Deck, the implicit numerical solver is used in place of the explicit solver, and the time stepping as summarized in Table 2. Note that, the majority of spectral wave models

utilize a $1_{st}$ order time-space method to solve the Wave Action Equation (WAE). Implicit schemes are constrained to $1_{st}$ order time-space methods due to the Godunov Theorem, limiting higher-order accuracy to nonlinear approaches and leading non-monotonic methods to produce negative wave action and highly dispersive results with respect to the time step (Booij et al., 1999). Notably, experiments employing implicit schemes in nearshore environments have revealed a significant lag between the physical time scale variation and the stability time scales governed by explicit schemes (CFL), justifying the quasi-steady

temporal variation in physical variables and validating the use of a $1_{st}$ order time-stepping scheme. Janenko (1971) supports this approach due to the temporal scale mismatch between physical and stability time steps for application of implicit schemes to hyperbolic problems. Janssen (2007) argues against employing higher-order methods in geographical space, suggesting that adding numerical diffusion to counteract the Garden Sprinkler Effect (GSE) degrades the solution and only affects distantly advected low-frequency waves (swell), such as those traveling from the Arctic to the Indian Ocean, a concern irrelevant to

the current study. Consequently, our work establishes the groundwork for furthering higher-order methods by employing a 1st order implicit scheme, representing the current state of scientific understanding in fully monotone integration of the WAE.

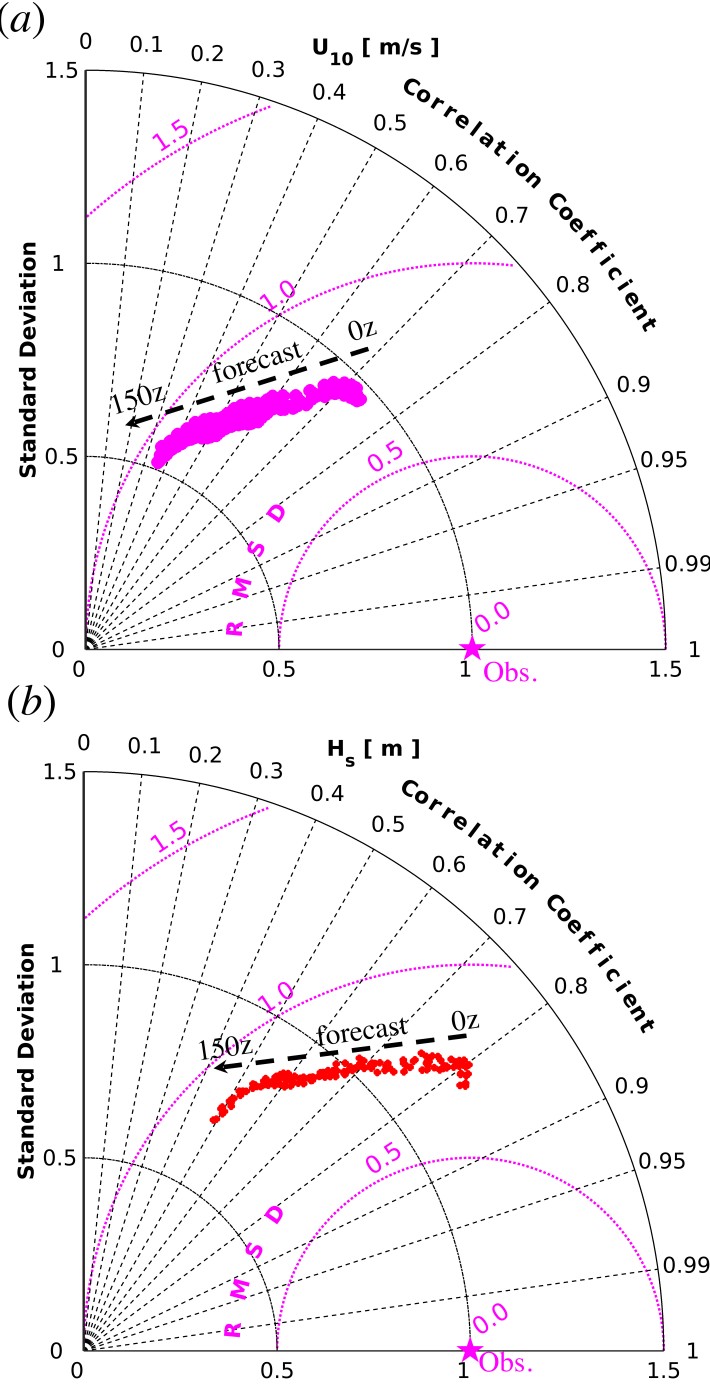

**Figure 8.** Taylor diagram for (a) wind speed ($U_{10}$) and (b) significant wave height ($H_s$) in terms of the Pearson correlation coefficient, normalized root mean square deviation ($RMSD$), and standard deviation $\sigma$. The black arrow shows the statistics in forecast hour (for June-July 2022).

| Solver | $\Delta t$ (s) | | | |
|---|---|---|---|---|
| | **Global** | **Spatial** | **Spectral** | **Source term** |
| **Explicit** | 180 | 60 | 90 | 10 |
| **Implicit** | 600 | | | |

**Table 2.** Model time steps for GLWUv2.0 with the explicit solver and experimental study with the implicit solver.

A validation study is performed on three unstructured meshes ($G0$, $G1$, and $G2$) for ten stormy conditions (T1,...,T10) in the Great Lakes from June 27 to October 26, 2016. The atmospheric forcing is provided by the HWRF model (Tallapragada et al., 2015) covering Lake Michigan and Lake Superior. The $G0$ mesh, which is used in the operational GLWUv1.0 and GLWUv2.0 systems, is compared with two other meshes of higher resolutions. The top row of Figure 9 shows the grid resolutions with an inset magnified view on the northeast side of Lake Michigan. The meshes are finer near the shore and at locations with irregular bathymetry to ensure the complicated coastline and geometries, which have a significant influence on wave transformations, are better represented in the model.

A snapshot of the significant wave height field on the $G0$ mesh and the percentage difference for $G1$ and $G2$ meshes are shown in the bottom row. In the bottom left-hand panel, the significant wave height is shown, which is extracted from simulations on the $G0$ grid. The middle panel illustrates the percentage changes between the $G0$-$G1$ grids, while the right panel shows the percentage changes between the $G1$-$G2$ grids. These changes indicate approximately a $5\%$ variation in the domain extent. These variations primarily occur in regions characterized by sharp gradients in bathymetry, where the higher-resolution meshes can effectively resolve the terrain with a sufficient number of elements. In addition to a quantitative comparison, the atmospheric and wave model outputs are validated at six available stationary buoy stations. The performance of the atmospheric and wave models at the six buoy locations are summarized in Taylor diagrams shown in Fig. 10, in terms of normalized standard deviation ($\sigma$), root mean square deviation ($RMSD$), and correlation coefficient ($CC$) for the observation and model outputs for wind speed (top panel), significant wave height (middle panel), and peak period (bottom panel).

The wind speed has normalized standard deviations in the range of 1-1.3, $RMSD$ between 0.5-0.8 with a correlation coefficient of nearly 0.8 for all the buoys. On the other hand and for all the grids, the significant wave height has $\sigma = 1 - 1.3$, $RMSD = 0.42 - 0.6$, and $CC = 0.86 - 0.94$. The peak period has a similar normalized standard deviation, with $RMSD = 0.6 - 0.81$ and $CC = 0.75 - 0.88$. These results show a slight improvement for the intermediate ($G1$) and high-resolution grids ($G2$) with the implicit scheme relative to the operational grid ($G0$) with the explicit scheme. Note that a significant improvement is not expected due to the deepwater location of the available buoys, which are not expected to benefit from higher resolutions nearer the coast. Time series of wind speed $U_{10}$, wind direction (Fig.A1), significant wave height ($H_s$), peak period ($T_p$), and mean wave direction from buoy observation and HWRF/WW3 models in Fig.B1 are provided in supplementary materials. Unlike the forecast (section 1.4), where the upstream atmospheric model and downstream wave model diminish in accuracy with an increase in forecast lead time, the wind hindcast for 10 stormy conditions maintains its precision over time (Fig. A1). Consequently, the time series data for the significant wave height, peak period, and wave direction exhibit strong consistency

throughout the entire simulations (Fig. B1). As depicted in the middle panel of Fig. B1, the peak period of young waves during severe conditions remains below 8 seconds. Notably, in closed basins like the Great Lakes where swell generated in distant regions does not exist, the immediate influence of local wind on waves is more evident, showcasing the significance of the upstream wind model's accuracy in the behavior of the wave model.

Our findings highlight a noteworthy improvement in model output achieved by increasing the grid resolution, with particu-
larly remarkable enhancements observed in the nearshore region. The finer details of sub-scale geographical features that were previously absent in the coarse meshes are now captured in the higher-resolution simulations. This finding underscores the importance of higher-resolution grids in accurately representing coastal morphology features. These significant differences in model output hold the promise of qualitative improvements that can be of great importance for nearshore hazards forecasting and prediction. Forecasters in the region have a crucial need for such advancements to enhance their ability to predict and
mitigate potential coastal hazards effectively.

However, to observe a more pronounced impact of mesh resolution, especially for the ten stormy conditions studied, a comparable increase in atmospheric forcing resolution, coupled with changes in water level and current fields, becomes essential. The reason behind this requirement is that the wave climate in enclosed basins is primarily influenced by locally generated wind-seas. During this study, the same wind field was interpolated on all three grids without considering water levels and
current fields. In addition, due to the lack of coastal observations, where the dominant waves might interact with the bottom, the statistics of the three simulations show nearly equal performances. Looking solely at the comparison plots with the existing in-situ observations underestimates the importance of high-resolution bathymetic features because the gauges are far away from any geographical feature like offshore islands that would show the resolution effect on wave transformation interacting with those features (see conclusion remarks for the needs for the future improvements on high-resolution grids).
The model performance has been evaluated on NCEP's HPC's for the pre-existing parallelization algorithm in WW3 ($CD$) with its explicit equation solver and the newer Domain Decomposition ($DD$) algorithm with the implicit solver on three unstructured triangular meshes. The grid resolutions are compared in Fig. 9. The $G0$ mesh is designed for operational implementation. The criteria for the $G0$ mesh design were mainly computational efficiency and limited available resources in view of the requirements of the $CD$ parallelization and explicit propagation schemes. Therefore, the $G0$ grid is relatively small with
nearly $250K$ nodes and a minimum resolution of 250 m in coastal areas. In contrast, a moderate ($G1$ with $750K$ nodes) and a large grid ($G2$ with $2.40M$ nodes) with $\sim 5$ m minimum resolution were designed to distinguish the computational limits of each parallelization algorithm and solver scheme.

The scalability performance is shown in Fig. 11 in terms of non-dimensional computational speed, revealing linear growth in the model performance for various model options and grids for implicit (solid lines) and explicit (dashed lines) schemes. It is
clearly shown that the $G0$ grid (black lines) has one order of magnitude faster performance with the implicit scheme compared to the explicit scheme and faster than real-time computation for any given number of CPU cores for both solver schemes. However, increasing the number of grid nodes to 750 k and decreasing the minimum resolution to 5 m led to a significant slowdown in the model performance for the explicit solvers (dashed blue) due to the ineffectiveness of $CD$ parallelization and model CFL constraint embedded in the explicit scheme on triangular unstructured grids.

Such limitations have confronted WW3 users applying larger grids with very high resolution for a long time. The shown slowdown in performance is more evident for the $G2$ mesh (dashed red), where the model computational wall time dropped under real-time (horizontal solid line). On the other hand, the implicit scheme (solid lines), allows resolution of the physical processes in nearshore regions with the prescribed higher grid resolution in an efficient way. For example, the $G2$ mesh with the $DD$ scheme has nearly the same performance of $G0$ mesh with the explicit scheme, in spite of having nearly ten times

more nodes. In addition, the $DD$ algorithm does not have any limit in terms of the number of CPU cores allocations, unlike the $CD$ algorithm. Therefore, larger grids can be distributed on a larger number of computational cores, free from the NSPEC limit (number of spectral components) imposed in the $CD$ algorithm (vertical solid line). This computational performance breakthrough is a valuable contribution from our work to research and operational applications using the WW3 model.

## 3 Conclusions

This article presents an overview of the implementation of GLWUv2.0, encompassing workflow design and validation studies for the duration of 4 months re-forecast, so-called retro-respective simulations. The validation includes a qualitative comparison during the summer season and a qualitative analysis for the ice season when no buoys were available in the Great Lakes. The article also acknowledges the limitations and challenges encountered in the operational environment, which restricted the utilization of the most advanced components of the wave model. Model tuning focused on minimizing statistical metrics for

significant wave height, aligning initial forecasts closely with observations. However, as forecast lead time increased, the model's accuracy decreased, due to higher uncertainty in the forcing field. The relationship between wind and wave model outputs showed discrepancies, with wind being consistently overestimated for smaller wind speeds and underestimated for larger wind speeds as the forecast progressed. The impact of forecast lead time on NDFD winds and WW3 waves was analyzed through Taylor diagrams, indicating a decrease in accuracy over time for both wind and wave forecasts in terms of deviation

and correlation coefficient.

In addition to GLWUv2.0 implementations, the model scalability was evaluated on three unstructured meshes, with $250K$ ($G0$), $750k$ ($G1$), and $2.4M$ ($G2$) nodes with minimum resolutions of 250, 20, and 5 m, respectively. The higher-resolution meshes show a 5% variation in domain extent, primarily resolving sharp bathymetric gradients more effectively. The comparison at buoy stations reveals slight improvements in wave prediction for higher-resolution grids. However, significant improve-

ments were not expected due to buoy locations being distant from coastlines. The scalability analysis shows the efficiency of the implicit numerical solver as opposed to the explicit solver, constrained by CFL criteria. In addition, the limitless Domain Decomposition ($DD$) parallelization will support the use of large meshes in operational environments, where simulation time mandates the design process. In order to show the accuracy of the model during storm conditions, the aforementioned unstructured meshes were compared during ten selected stormy conditions.

The article explores studies that offer insights into future improvements in wave forecasting. It emphasizes the need for refining grid resolution and modernizing numerical methods to achieve significant enhancements. Additionally, considering coastal-scale phenomena, interactions with other earth system models such as circulation, sea-ice, atmosphere, and hydro-

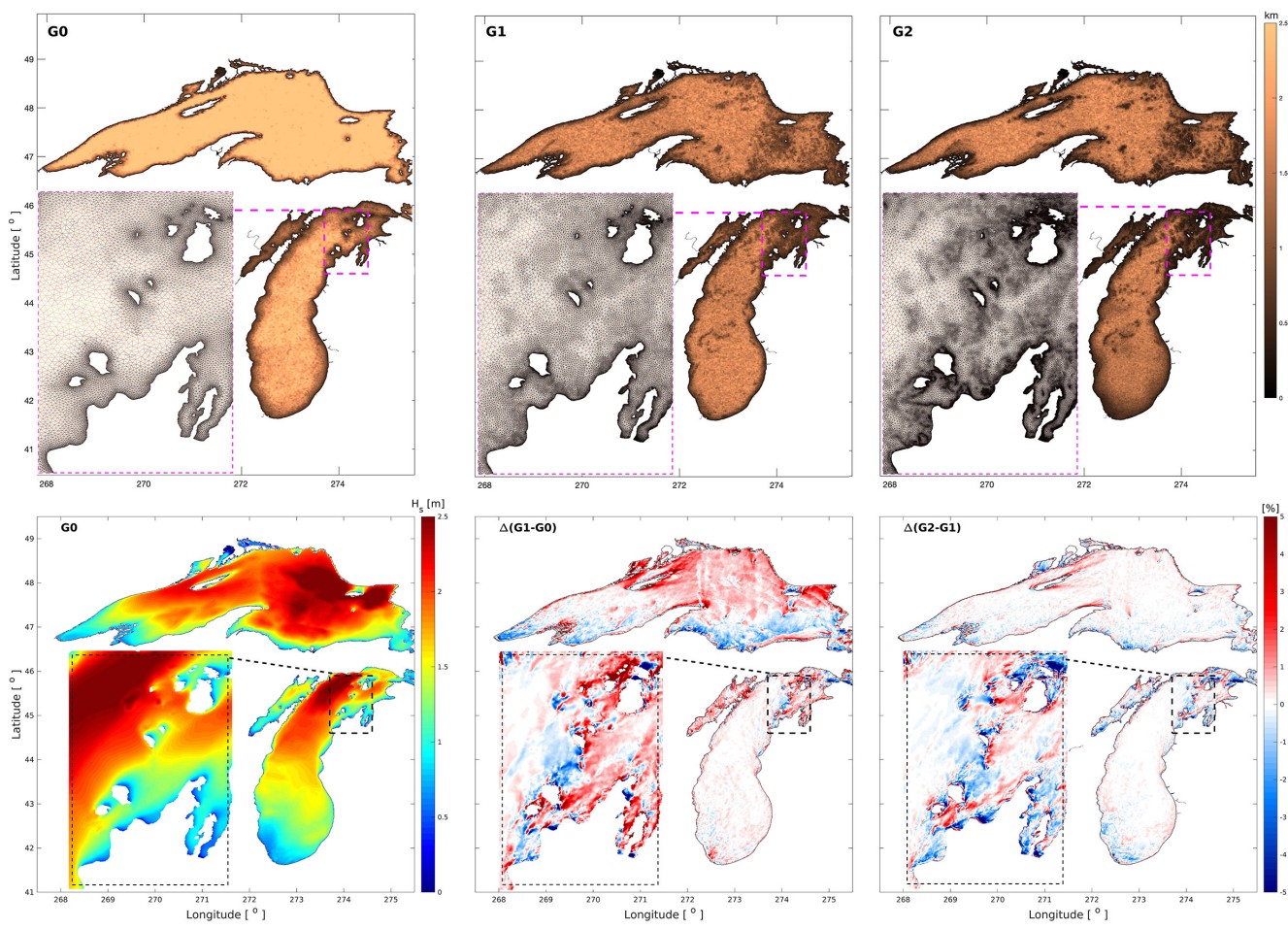

**Figure 9.** Top row: Grid resolution for Lake Michigan and Lake Superior with a closer view at the northeast side of Lake Michigan for mesh $G0$ ($\sim 250K$), $G1$ ($\sim 750K$) and $G2$ ($\sim 2.4M$); Bottom row: A snapshot of significant wave height on mesh $G0$ (left); percentage change in $H_s$ field between $G1$ and $G0$ (middle); and $G2$ and $G1$ (right).

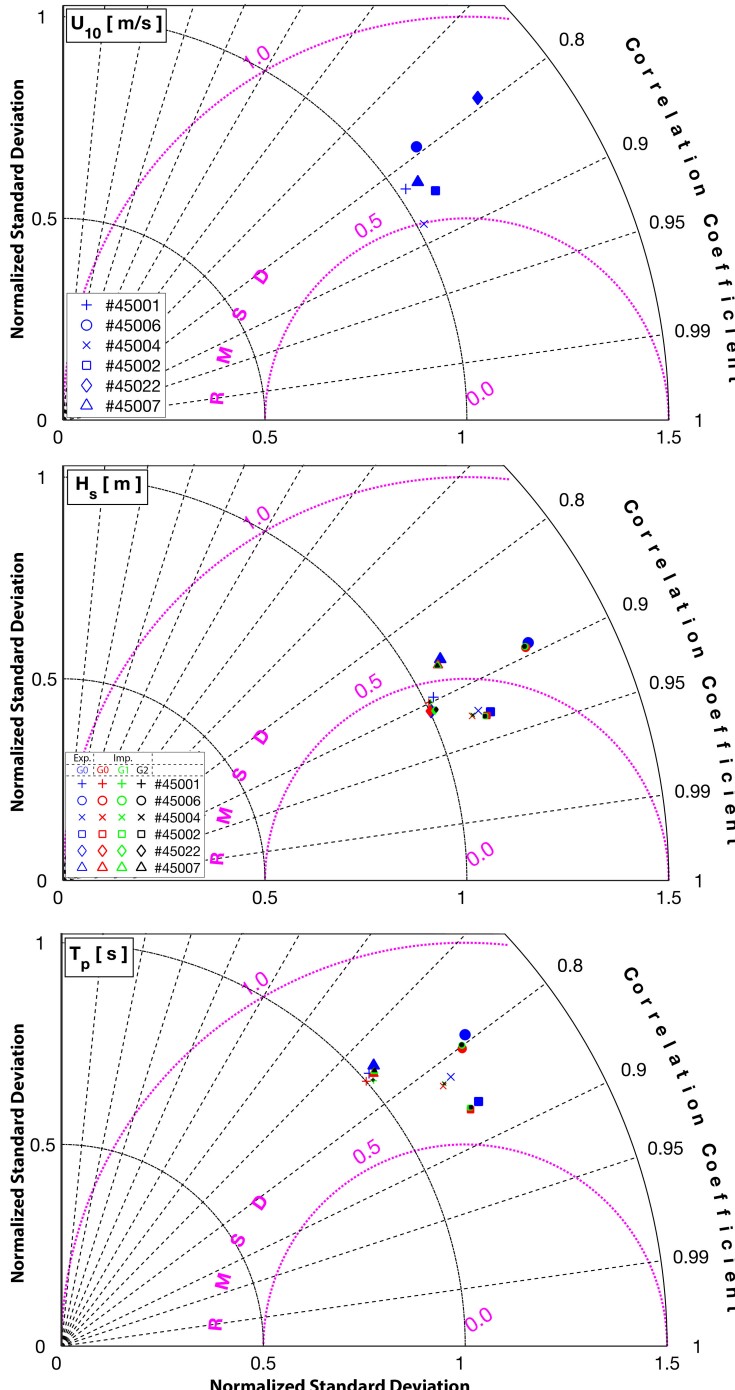

**Figure 10.** Taylor diagram for wind speed ($U_{10}$: top); significant wave height ($H_s$: middle); and peak period ($T_p$: bottom) representing modeled and collected data at buoy locations (blue: Explicit on mesh $G0$; red: Implicit on Mesh $G0$; green: Implicit on Mesh $G1$; and black: Implicit on Mesh $G2$) in terms of the Pearson correlation coefficient ($CC$), root mean square deviation ($RMSD$), and normalized standard deviation $\sigma$.

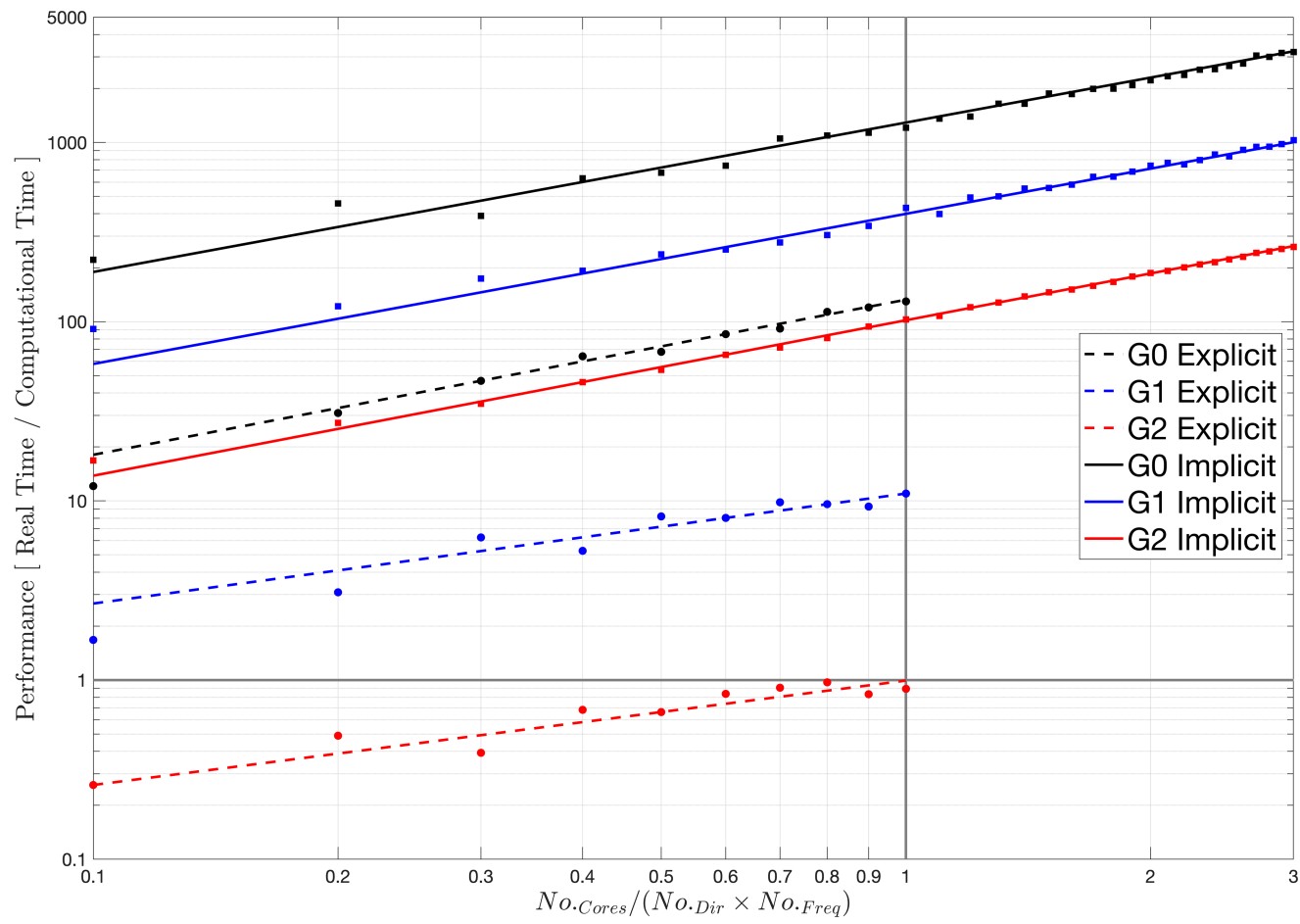

**Figure 11.** Model performance (v7) on HPC environment and scalability of WW3 models for explicit numerical solver with Card Deck parallelization (dashed lines) and implicit scheme with Domain Decomposition parallelization algorithm (solid lines) for three unstructured grids with $250K$ ($G0$), $750K$ ($G1$) and $2.4M$ ($G2$) nodes. The horizontal axes show the number of computational cores normalized by the number of frequencies and directions. The vertical solid line represents the $CD$ limit ($No.Cores = No.Dir \times No.Freq$). The horizontal solid line represents real-time performance. The tests are performed on NCEP's HPC, Hera, equipped with 2.60 GHz Intel Haswell CPU and 2.67 GB memory/core.

logical models is crucial for a comprehensive understanding of coastal dynamics. The role of coastal observations in model validation is highlighted, requiring remote sensing techniques such as coastal altimetry and new in-situ data collection technolo-

gies such as drifter buoys and HF radars, particularly during the ice season when conventional field observation is not possible. Improving our understanding of the wave climate in diverse environments (i.e., interacting with ice, vegetation, rocky beaches, and coastal structures) through more observations and with more accurate numerical models will help to better evaluate and design coastal protection with nature-based solutions, so-called Engineering with Nature, to improve coastal resilience.

In addition to the wave model core upgrade, GLWUv2.0 features seven additional field outputs in order to be utilized in

the Dangerous Seas Project over the Great Lakes Region. This project is a bilateral collaborative agreement between NOAA (EMC-OPC) and the Environment and Climate Change Canada (ECCC), to enhance scientific and operational cooperation between Canada and the United States. In this effort, a Dangerous Sea is defined as a combination of wave height, period, steepness, breaking waves, crossing seas, and rapidly changing sea state (time rate of change) that causes navigational risk (speed and course) and/or leading to, the potential loss of vessel, cargo, or crew. Mariners should avoid dangerous seas at all

costs. The project has started with the Great Lakes, a basin responding to mostly the wind impact on the water surface, and will migrate the work to the open ocean where wave patterns are more complex. Additional details on this will be made available in a separate article in the future.

*Code availability.* The current version of GLWUv2.0 modeling system, including the workflow, the WW3 model, and input files to produce the results, shown in this paper, can be accessed at the Zenodo archive: https://doi.org/10.5281/zenodo.8341987 under the Lesser GNU Public

License v3.

*Author contributions.* AA: conceptualization, methodology, WW3 code development, workflow development and implementation, data curation, visualization, validation, writing—original draft; SB: methodology, workflow development and implementation, data curation, visualization, validation, writing—review, and editing; JHA: WW3 code development, workflow development and implementation, data curation, writing—review, and editing; AR: WW3 code development, writing—review, and editing. TJH: WW3 code development, writing—review,

and editing; MAB: WW3 code development, writing—review, and editing; JMS: WW3 code development, writing—review, and editing.

*Competing interests.* The authors declare no competing interests.

*Acknowledgements.* The author wishes to thank Dr. Robert E. Jensen for the fruitful discussions on the future of wave modeling/observations in the Great Lake area. The authors acknowledge the NOAA's Great Lakes Environmental Research Laboratory (GLERL) for providing the bathymetry and coastline of the Lake Champlain and buoy #45195 observations, Dr. Greg Mann from the National Weather Service, Detroit

office for providing 10 storm cases, the forecast officers at the Great Lakes WFOs for their feedback during the evaluation period, particularly

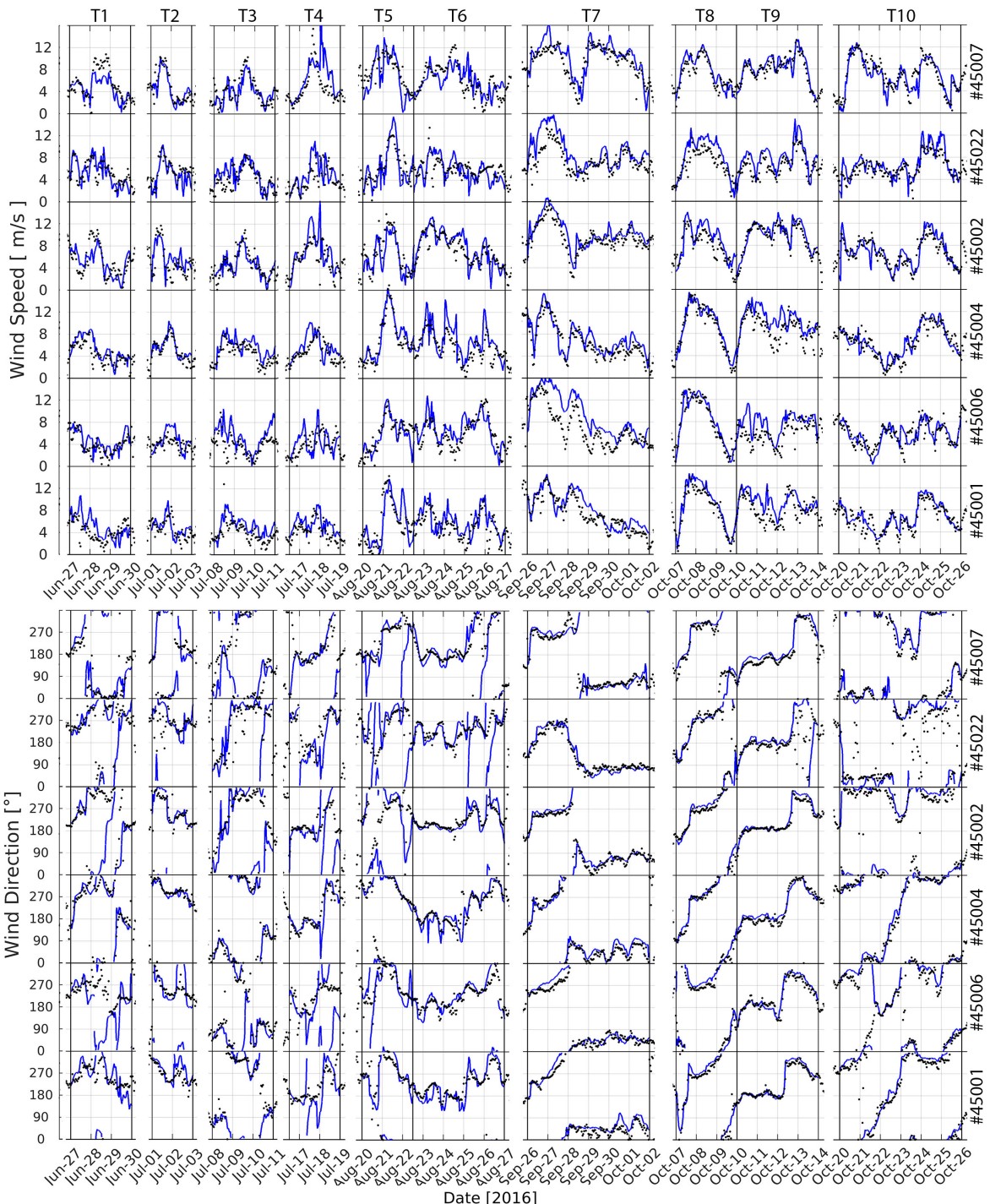

**Figure A1.** Atmospheric Model validation at the buoy locations, HWRF (blue) versus observations (black): (top) Wind speed; (bottom) Wind direction.

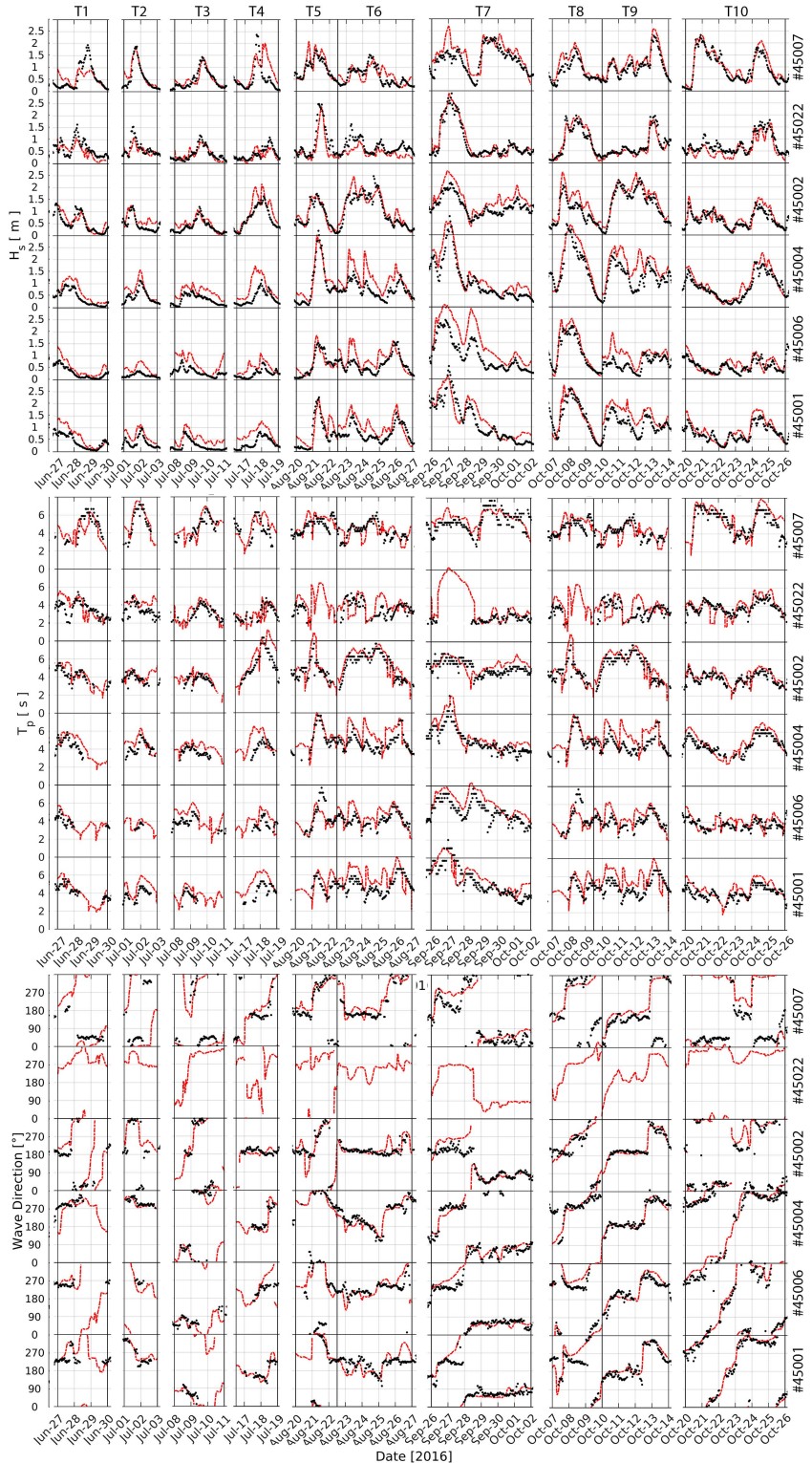

**Figure B1.** Wave Model validation at the buoy locations, WW3 (red) versus observations (black): (top) Significant wave height; (middle) Peak Period; (bottom) Wind direction.

during ice season, when eyewitnesses were the only source of validation, EMC Engineering and Implementation Branch and NCEP Central Operation (NCO) for their help during GLWUv2.0 transition to operations.

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
