# Peer review of "Great Lakes Waves Forecast System on High-Resolution Unstructured Meshes"

_Geoscientific Model Development, 2023_

## Author Comment (AC1)

**Overview:**

In this paper "Great Lakes Waves Forecast System on High-Resolution Unstructured Meshes" by Abdolali and coauthors, the Great Lakes Wave Forecast system is described. Its evolution from first-generation parametric model to a third generation full spectral model with an unstructured mesh (GLWUv2.0). The operational implementation of the GLWUv2.0 is described: its meshes, forcing fields and workflow. A validation was done comparing the operational model output against in-situ measurements from 25 buoys. The numerical system was also run using some key features that WAVEWATCH III, which is the core numerical model, already has and will be used in future operational implementation of the wave forecast system. The output for the later runs was compare to measurements from 6 buoys.

We are very grateful to the reviewer for his/her constructive critiques and comments. In the following, we state the referee's comments (in blue) followed by the response and actions taken (in black).

**General recommendation:**

The paper has a value for the wave modeling community, it is well structured. However there is a lack of connection between objectives, the proposed experiments, the results and their conclusions. The paper needs a major revision. I recommend to revise the paper and resubmit it.

**Major comments:**

The abstract should include briefly the key conclusions of the study.

We expanded the abstract to highlight two additional conclusion remarks:

1- With the recent development, there are no limits in terms of computational scalability and minimum resolution in the coastal areas.

2- The GLWU system's performance showcased its adeptness in predicting waves accurately at the start of the forecast when the atmospheric model is precise, as well as throughout the entire hindcast for stormy conditions. In closed Great Lakes basins untouched by the lateral swell, the atmospheric model's direct impact on wave behavior stands apart, showing reduced forecast accuracy over time, while maintaining consistent precision in accurately wind-hindcasted stormy conditions.

For numerical wave modelling on high geographical-resolution and shallow waters, highquality bathymetry is essential, however there is a lack of information about the source of the bathymetry and its quality. This must be addressed since in some numerical experiments the wave system has a resolution up to 5 meters.

We added the following to the manuscript as reference to the DEM, used for mesh generation:

The Great Lakes Bathymetry collection compiles geological and geophysical data of lake floors, including bathymetry and detailed maps sourced from over a century of soundings by various organizations like the U.S. Army Corp. of Engineers, NOAA, and the Canadian Hydrographic Service. NCEI/NOAA compiled unified topobathy data for Lake Erie and Saint Claire (NGDC, 1999a), Huron (NGDC, 1999b), Michigan (NGDC, 1996), Ontario (NGDC, 1999c) and Superior and provided public access to this data.

The topobathymetric grid for the generation of the Lake Champlain mesh was obtained by refining the Environment and Climate Change Canada (ECCC)-developed mesh, integrating data from 15 sources for a detailed two-dimensional hydrodynamic model. Covering from South Bay to the Poultney River in Whitehall, NY, and extending northward to Fryers Rapids near St Jean, QC, it intricately maps 14 significant river inputs to the lake. This grid encompasses surrounding floodplains to simulate various inundation scenarios across the spectrum of water level fluctuations experienced within the region (Titze et al. 2023).

References:

National Geophysical Data Center, 1999. Bathymetry of Lake Erie and Lake St. Clair. National Geophysical Data Center, NOAA. doi:10.7289/V5KS6PHK

National Geophysical Data Center, 1999. Bathymetry of Lake Huron. National Geophysical Data Center, NOAA. doi:10.7289/V5G15XS5

National Geophysical Data Center, 1996. Bathymetry of Lake Michigan. National Geophysical Data Center, NOAA. doi:10.7289/V5B85627

National Geophysical Data Center, 1999. Bathymetry of Lake Ontario. National Geophysical Data Center, NOAA. doi:10.7289/V56H4FBH

Titze, D., Beletsky, D., Feyen, J. et al. Development and skill assessment of a real-time hydrologic-hydrodynamic-wave modeling system for Lake Champlain flood forecasting. Ocean Dynamics 73, 231–248 (2023). https://doi.org/10.1007/s10236-023-01550-2

Section 2.5 is totally disconnected from the rest of the article. "Dangerous Seas", this topic does not appear in any other section; from introduction to conclusions. There are not

numerical experiments presented in this paper related to this topic. It could be mentioned in a couple of lines in "Future Implementation" section, but it doesn't deserve a full subsection.

We moved this section to the conclusion and it is no longer a subsection.

The Conclusions section offer, again, a description of the system, description of the unstructured meshes, describe the need for high resolution meshes and coupling with other Earth systems, etc. It does not offer conclusion of all the implementation, validation and statistics done.

We added the following to address reviewer's comment:

For forecast:

Model tuning focused on minimizing statistical metrics for significant wave height, aligning initial forecasts closely with observations. However, as forecast lead time increased, the model's accuracy decreased, due to higher uncertainty in the forcing field. The relationship between wind and wave model outputs showed discrepancies, with wind being consistently overestimated for smaller values and underestimated for larger values as the forecast progressed. The impact of forecast lead time on NDFD winds and WW3 waves was analyzed through Taylor diagrams, indicating a decrease in accuracy over time for both wind and wave forecasts in terms of deviation and correlation coefficient.

For hindcast:

The higher-resolution meshes show a 5% variation in domain extent, primarily resolving sharp bathymetric gradients more effectively. The comparison at buoy stations reveals slight improvements in wave prediction for higher-resolution grids. However, significant improvements were not expected due to buoy locations being distant from coastlines.

The values of statistical parameters, when comparing $G0$, $G1$ and $G2$, do not show a "noteworthy improvement" as it is stated (line #209), as a matter of fact the authors declares that "due to the lack of coastal observations (line #220) ... the three simulations show nearly equal performance (line #221)".

We agree with the reviewer's comment on the minimal impacts of mesh resolution at existing buoy locations. We then discussed this effect in the field plot (Figure 9) and added the following to describe it:

A snapshot of the significant wave height field on the $G0$ mesh and the percentage difference for $G1$ and $G2$ meshes are shown in the bottom row. In the bottom left-hand panel, the

significant wave height is shown, which is extracted from simulations on the $G0$ grid. The middle panel illustrates the percentage changes between the $G0$-$G1$ grids, while the right panel shows the percentage changes between the $G1$-$G2$ grids. These changes indicate approximately a 5% variation in the domain extent. These variations primarily occur in regions characterized by sharp gradients in bathymetry, where the higher-resolution meshes can effectively resolve the terrain with a sufficient number of elements.

The implicit scheme makes the forecast system to finish faster but the price we have to pay is the numerical diffusion. Something should be mentioned about this topic and I guess a time step must be provided when using the $DD$ scheme. What were the time steps used?

Currently, the majority of spectral wave models utilize a $1_{st}$ order time-space method to solve the Wave Action Equation (WAE). Implicit schemes are constrained to $1_{st}$ order time-space methods due to the Godunov Theorem, limiting higher-order accuracy to non-linear approaches and leading non-monotonic methods to produce negative wave action and highly dispersive results with respect to the time step (Booij et al., 1999). Notably, experiments employing implicit schemes in nearshore environments have revealed a significant lag between the physical time scale variation and the stability time scales governed by explicit schemes (CFL), justifying the quasi-steady temporal variation in physical variables and validating the use of a $1_{st}$ order time-stepping scheme. Yanenko (1971) supports this approach due to the temporal scale mismatch between physical and stability time steps for application of implicit schemes to hyperbolic problems. Janssen (2008) argues against employing higher-order methods in geographical space, suggesting that adding numerical diffusion to counteract the Garden Sprinkler Effect (GSE) degrades the solution and only affects distantly advected low-frequency waves (swell), such as those traveling from the Arctic to the Indian Ocean, a concern irrelevant to the current study. Consequently, our work establishes the groundwork for furthering higher-order methods by employing a 1st order implicit scheme, representing the current state of scientific understanding in fully monotone integration of the WAE.

Booij, N. R. R. C., Roeland C. Ris, and Leo H. Holthuijsen. A third-generation wave model for coastal regions: 1. Model description and validation. Journal of geophysical research: Oceans 104.C4 (1999): 7649-7666.

Janenko, N. N. (1971). The method of fractional steps (Vol. 160). New York: Springer. Janssen, Peter AEM. Progress in ocean wave forecasting. Journal of Computational Physics 227.7 (2008): 3572-3594.

The time steps used for the simulations are summarized in the following table and added to the manuscript (table 1).

The authors mentioned that for the winter simulations, there were not buoys measuring waves and "...only qualitative checks were performed". Those checks are not shown in the paper. How good or bad the forecast system was qualitatively?

| | $\Delta t$ (s) | | | |
|---|---|---|---|---|
| **Solver** | **Global** | **Spatial** | **Spectral** | **Source term** |
| **Explicit** | 180 | 60 | 90 | 10 |
| **Implicit** | 600 | | | |

Table 1: Model time steps for GLWUv2.0 with the explicit solver and experimental study with the implicit solver.

Following the reviewer's comment, we added a field plot and animation (supplementary) during ice season, showing a reasonable wave field when ice is present in the domain. Note that due to the unavailability of observations during the ice season, and before the implementation of the GLWUv2.0 system in operation, we collected feedback from the Weather Forecast Offices (WFOs). Based on their visual observations, the model outputs are in good agreement with their observations from the field. We also tested the GLWUv2.0 against the GLWUv1.1 where we had wave artifacts at the edges of ice fields, reported by WFOs. The comparison plots are attached here, showing the mitigation of wave artifacts in the model outputs.

Lines 207-208. Figures A1 and B1 are mentioned but the results were not described nor discussed and they were not used to conclude anything. Any description or conclusion is left to the reader. In this case those figures do not add any value to the paper.

We added additional information about the time series shown in supplementary figures to address the comment, provided by the referee. Note that these two figures are in the supplementary section, to support the statistics shown in Taylor diagrams:

Unlike the forecast (section 2.4), where the upstream atmospheric model and downstream wave model diminish in accuracy with an increase in forecast lead time, the wind hindcast for 10 stormy conditions maintains its precision over time (Fig. A1) Consequently, the time series data for the significant wave height, peak period, and wave direction exhibit strong consistency throughout the entire simulations (Figure B1). As depicted in the middle panel of Figure B1, the peak period of young waves during severe conditions remains below 8 seconds. Notably, in closed basins like the Great Lakes where lateral sea swell does not impact, the immediate influence of local wind on waves is more evident, showcasing the significance of the upstream wind model's accuracy in the behavior of the wave model

**Minor comments:**

Line 3. Instead of "is successfully tackle by" could be "is successfully tackle in part". There is a need for implementation of more accurate physics, as it is mentioned in lines 37-8.

Corrected in the manuscript.

[Figure]

Figure 1: Comparison of the GLWU domain significant wave height between the GLWUv1.1 (a) and the development GLWUv2.0 (b) during ice season (a snapshot on February 3rd, 2022 12:00:00 EST). The green circles in panel (a) show the wave artifacts along ice edges.

Line 8. "Our results describe the development..." The results section should not be used to describe the development of the wave forecast system.

Corrected in the manuscript.

Lines 13-14. In the US population living in the Great Lakes region the entire states population is taking into account, however for Canada the population is taking only as a part of Ontario, please review the literature on how many people lives in Ontario, and set the percentage related to Canada, as it was for the US.

The info is added to the manuscript.

Lines 31-32. "Two years later in 2006", to years later compared to what? There is not a reference to the year 2004.

Added the reference to the year 2004 in the manuscript.

Line 50. "allowing very large meshes", $CD$ allows very large meshes as well, but what is the difference? "allowing very large meshes to run in short time"?

We clarified the following in the manuscript:

The contrast lies in how Domain Decomposition ($DD$) surpasses Card-Deck ($CD$) concerning scalability with a large number of CPUs. $CD$ has a restricted maximum count of CPUs compared to the unlimited count in $DD$. Moreover, when dealing with finer resolution meshes, the implicit solver in $DD$ enables operation with larger CFL numbers, whereas the explicit solver in $CD$ is limited by $CFL < 1$, resulting in slower model performance.

Line 66. "Section 3" should be Section 2".

Corrected in the manuscript.

Line 81. "The WW3 model" should be "The GLWUv2.0", as WW3 can have a different values for the parameters, but the values provided there are specifically for the Forecast System.

Corrected in the manuscript.

Line 83. Need space between et al. and (2010).

Corrected in the manuscript.

 "a stationary ice concentration at the initialization time step", then, what is provided after the initial time step? A non-stationary ice concentration? The ice field is keep constant in time or there is a forecast system for the ice concentration?

Clarified in the manuscript: The ice concentration is Stationary, defined at the initialization time step and kept constant for the entire cycle.

Lines 109-110. A resolution for the HRRR winds is provided but no for GFS winds.

Added to the manuscript.

Line 117. No need to repeat the list of the Great Lakes.

Removed from the manuscript.

Line 123. "In case the current cycle is not available" should be "In case the forcing for the current cycle is not available".

Corrected in the manuscript.

Line 124. "If the ice field is not provided, the previous forecast cycle ice field is used" So, is there a forecast system that provides forecasted ice fields? Or are those analyzed fields which are provided by NIC and they are kept constant in time for the whole forecast window? This is not clear.

Clarified in the manuscript.

Line 139. What is the running time for the long (or short) cycle for Lake Champlain?

Added to the manuscript.

Line 149. "25 locations, shown" instead of "25 locations as shown".

Corrected in the manuscript.

Line 151. "Which was one of the criteria", where there other criteria used? Which ones?

Corrected in the manuscript.

Figure 10. In the caption, instead of "normalized by frequency and directional resolution" should be "normalized by the number of frequencies and directions" as indicated in the x-axis.

Corrected in the manuscript.

---

## Author Comment (AC2)

The paper "Great Lakes Waves Forecast System on High-Resolution Unstructured Meshes" by Abdolali et al presents the Great Lakes Wave Forecast System, from the history of the first models in the area, to the present operational system and the future developments. The paper is well structured and written. However, sections are quite concise as the topics treated are often just mentioned. Also, the paper deals with a complex forecasting system in a complex geographical environment, which would require more detailed descriptions of the different elements involved (the forcings, the details of the modelling workflow, just to mention a couple) and of the figure and results presented. My recommendation is to revise the paper expanding the presentation, in particular of the topics of Section 2.

We are very grateful to the reviewer for his/her constructive critiques and comments. In the following, we state the referee's comments (in blue) followed by the response and actions taken (in black).

**Specific comments:**

line 77: the forecasting schedule is not clearly enough explained in my opinion, please make it more clear to the reader

We clarified it in the manuscript:

The Great Lakes wave forecast operates on an hourly basis, incorporating both short and long cycles, as depicted in Figure 3. There are twenty short cycles, each running for 48 forecast hours. Additionally, there are four long cycles, which extend for 150 forecast hours and are scheduled at 01z, 07z, 13z, and 19z.

line 94: it would be useful if authors repeat here the resolution of the Great Lakes grid

We added the following table and modified the text to address the comment:

Mesh resolution and corresponding histograms, highlighting the distribution of element size and significance of coastal elements in comparison to deep-water elements, are shown in Fig. 1. and the summary is provided in table 1.

Figure 2: acronyms should be added in the caption; also, details of the forcings are quite concise and might be better explained

The caption of Fig. 2 is modified:

Great Lake Wave Unstructured v2.0 atmospheric and ice forcing hierarchy. For wind, the National Digital Forecast Database (NDFD) and a combination of High-Resolution Rapid

| Lake | # Node | # Element | $\Delta x_{min}$ (m) | $\Delta x_{max}$ (m) |
|---|---|---|---|---|
| Superior | 51k | 81k | 246 | 3300 |
| Michigan | 58k | 103k | 250 | 2470 |
| Huron | 64k | 101k | 203 | 2840 |
| Erie | 45k | 78k | 203 | 1603 |
| Ontario | 35k | 57k | 224 | 2150 |
| Champlain | 30k | 60k | 60 | 400 |

Table 1: Mesh characteristics for Lakes Superior, Michigan, Huron, Erie, Ontario and Champlain in terms of number of nodes and elements, minimum and maximum resolutions.

Refresh (HRRR) and Global Forecast System (GFSv16) are used for the five lakes and Lake Champlain, respectively. The ice is taken from the National Ice Center (NIC) and WFO Burlington help-desk for the five lakes and Lake Champlain, respectively.

We added the following in the text to describe the missing info about NDFD, HRRR and GFS:

The National Digital Forecast Database (NDFD) is a combination of data from regional NWS Weather Forecast Offices (WFOs) and the National Centers for Environmental Prediction (NCEP) models (Glahn and Ruth, 2003). The Global Forecast System (GFSv16) (NOAA, 2021) from the National Centers for Environmental Prediction (NCEP) serves as a fundamental component in NCEP's operational numerical guidance suite for global climate modeling. It offers both deterministic and probabilistic global forecasts, extending up to 16 days, and plays a key role by supplying initial and boundary conditions for NCEP's regional, ocean, and wave prediction models.
The High-Resolution Rapid Refresh (HRRR) (Dowell et al 2022) is a NOAA real-time 3-km resolution, hourly updated, cloud-resolving, convection-allowing atmospheric model, initialized by 3km grids with 3km radar assimilation. Radar data is assimilated in the HRRR every 15 min over a 1-h period adding further detail to that provided by the hourly data assimilation from the 13km radar-enhanced Rapid Refresh.
The National Ice Center (NIC) Data for the Great Lakes are created from daily ice analysis. The files contain information on ice conditions that are separated into total ice concentration, ice types with their respective concentrations, and ice floe size.

References:

NOAA, 2021: Upgrade NCEP Global Forecast Systems (GFS) to v16: Effective March 17, 2021. Service Change Notice 21-20, Updated. National Weather Service Headquarters, Silver Spring MD.

Dowell, D. C., and Coauthors, 2022: The High-Resolution Rapid Refresh (HRRR): An

hourly updating convection-allowing forecast model. Part I: Motivation and system description. Wea. Forecasting, https://doi.org/10.1175/WAF-D-21-0151.1.

Glahn, Harry R., and David P. Ruth. The new digital forecast database of the National Weather Service. Bulletin of the American Meteorological Society 84.2 (2003): 195-202.

Section 2.5: I do not see the reason for having a section without any result. This can be added in the conclusions or somewhere else

We moved it to the discussed section following reviewer's comment.

lines 205-215: the point the authors are raising here is quite important and they have already commented it well. However, I would stress a bit more the benefits of the high coastal resolution even in absence of a sea truth to prove them, by showing some examples.

In the manuscript, we discuss the example illustrated in Figure 8.

The second row in Figure 8 displays the differences between the $G0$ and $G1/G2$ grids. In the left-hand panel, the significant wave height is shown, which is extracted from simulations on the $G0$ grid. The middle panel illustrates the percentage changes between the $G0$-$G1$ grids, while the right panel shows the percentage changes between the $G1$-$G2$ grids. These changes indicate approximately a 5% variation in the domain extent. These variations primarily occur in regions characterized by sharp gradients in bathymetry, where the higher-resolution meshes can effectively resolve the terrain with a sufficient number of elements.

In the introduction a recent GLWU implementation which incorporates the implicit scheme and the current/water level forcing is mentioned. Then, the version 2.0 presently operational is presented, based on the explicit solver and no current/level forcing. And those two points are finally listed as future developments (the first is well elaborated, the second only mentioned). In my opinion, this generates confusion, so please, fix it.

We modified the text in the manuscript to clarify the research study, performed with the implicit solver and the operational implementation use of explicit.

---

## Author Response (AR2)

* * *
General recommendation The authors' responses to the first review are satisfactory, they have answered all questions and concerns. However, there are still some details that has to be addressed. The paper needs a minor revision. I recommend accepting the paper for publication once the following comments are addressed.

We are very grateful to the reviewer for his/her constructive critiques and comments. In the following, we state the referee's comments (in blue) followed by the response and actions taken (in black).

**Major comments:**

1. The Introduction section misses the section number and header. Because of this, all the section numbers and references to them are shifted.

Corrected.

2. The following paragraph shouldn't be included in the paper, for sure not in the Conclusions Section. Any sentence of such paragraph is not a conclusion of this study. "In addition to the wave model core upgrade, GLWUv2.0 features seven additional field outputs in order to be utilized in the Dangerous Seas Project over the Great Lakes Region. This project is a bilateral collaborative agreement between NOAA (EMC-OPC) and the Environment and Climate Change Canada (ECCC), to enhance scientific and operational cooperation between Canada and the United States. In this effort, a Dangerous Sea is defined as a combination of wave height, period, steepness, breaking waves, crossing seas, and rapidly changing sea state (time rate of change) that causes navigational risk (speed and course) and/or leading to, the potential loss of vessel, cargo, or crew. Mariners should avoid dangerous seas at all costs. The project has started with the Great Lakes, a basin responding to mostly the wind impact on the water surface, and will migrate the work to the open ocean where wave patterns are more complex. Additional details on this will be made available in a separate article in the future".
If anything, the first sentence "In addition to the wave model core upgrade, GLWUv2.0 features seven additional field outputs in order to be utilized in the Dangerous Seas Project over the Great Lakes Region." could be moved to section "Future implementation"

A new subsection is added to the end of manuscript and the above paragraph is moved to the "Future Development" section.

**Other comments**

1. Line 16. Instead of "U.S. states of" use a colon "U.S. states:"

Corrected.

2. Line 17. Instead of "two Canadian provinces of", use a colon "two Canadian provinces:"

Corrected.

3. Lines 147. It is stated that "For Lake Champlain, the process completes in 17 minutes in the short cycle and 27 minutes in the long cycle.", According to the running-times showed in lines 156, 162 and 169 should be "For Lake Champlain, the process completes in 16 minutes in the short cycle and 26 minutes in the long cycle."

Corrected.

4. Lines 186-188. The statistical parameters do not introduce less accuracy and more uncertainty in the forcing field.", replace "which introduced less accuracy and more uncertainty in the forcing field." by "which is introduced by the forcing field." Or other sentence that makes sense.

Corrected.

5. Line 205-206. Replace "new model features in meshes of one order of magnitude larger than" by "new model features in meshes that have one order of magnitude, in the number of nodes, larger than".

Corrected.

**comment by editorial office**

Notification to the authors: Regarding figure 4: with the next revision, please add the copyright icon as follows: © Google Earth.

Corrected.